# Nettle, a Long-Known Fiber Plant with New Perspectives

**DOI:** 10.3390/ma15124288

**Published:** 2022-06-17

**Authors:** Chloé Viotti, Katharina Albrecht, Stefano Amaducci, Paul Bardos, Coralie Bertheau, Damien Blaudez, Lea Bothe, David Cazaux, Andrea Ferrarini, Jason Govilas, Hans-Jörg Gusovius, Thomas Jeannin, Carsten Lühr, Jörg Müssig, Marcello Pilla, Vincent Placet, Markus Puschenreiter, Alice Tognacchini, Loïc Yung, Michel Chalot

**Affiliations:** 1UMR Chrono-Environnement, CNRS 6249, Université Bourgogne Franche-Comté, 25000 Besançon, France; chloe.viotti@univ-fcomte.fr (C.V.); coralie.bertheau-rossel@univ-fcomte.fr (C.B.); 2The Biological Materials Group, Department of Biomimetics, HSB—City University of Applied Sciences Bremen, Neustadtswall 30, 28199 Bremen, Germany; kathaalbrecht@web.de (K.A.); lea.bothe@hs-bremen.de (L.B.); jmuessig@bionik.hs-bremen.de (J.M.); 3Department of Sustainable Crop Production, Università Cattolica del Sacro Cuore, Via Emilia Parmense 84, 29122 Piacenza, Italy; stefano.amaducci@unicatt.it (S.A.); andrea.ferrarini@unicatt.it (A.F.); marcello.pilla@unicatt.it (M.P.); 4r3 Environmental Technology Ltd., Earley Gate, Reading RG6 6AT, UK; paul@r3environmental.co.uk; 5LIEC, CNRS, Université de Lorraine, 54000 Nancy, France; damien.blaudez@univ-lorraine.fr (D.B.); loic.yung@univ-lorraine.fr (L.Y.); 6Inovyn, 39500 Tavaux, France; david.cazaux@inovyn.com; 7Department of Applied Mechanics, FEMTO-ST Institute, Université Bourgogne Franche-Comté, 25000 Besançon, France; jason.govilas@femto-st.fr (J.G.); thomas.jeannin@femto-st.fr (T.J.); vincent.placet@univ-fcomte.fr (V.P.); 8Leibniz Institute for Agricultural Engineering and Bioeconomy (ATB), Max-Eyth-Allee 100, 14469 Potsdam, Germany; hjgusovius@atb-potsdam.de (H.-J.G.); cluehr@atb-potsdam.de (C.L.); 9Institute of Soil Research, University of Natural Resources and Life Sciences Vienna, 1180 Vienna, Austria; markus.puschenreiter@boku.ac.at (M.P.); alice.tognacchini@boku.ac.at (A.T.); 10Faculté des Sciences et Technologies, Université de Lorraine, 54000 Nancy, France

**Keywords:** *Urtica**dioica* L., stinging nettle, phylogeny, cultivation, fiber production and processing, phytomanagement

## Abstract

The stinging nettle *Urtica*
*dioica* L. is a perennial crop with low fertilizer and pesticide requirements, well adapted to a wide range of environmental conditions. It has been successfully grown in most European climatic zones while also promoting local flora and fauna diversity. The cultivation of nettle could help meet the strong increase in demand for raw materials based on plant fibers as a substitute for artificial fibers in sectors as diverse as the textile and automotive industries. In the present review, we present a historical perspective of selection, harvest, and fiber processing features where the state of the art of nettle varietal selection is detailed. A synthesis of the general knowledge about its biology, adaptability, and genetics constituents, highlighting gaps in our current knowledge on interactions with other organisms, is provided. We further addressed cultivation and processing features, putting a special emphasis on harvesting systems and fiber extraction processes to improve fiber yield and quality. Various uses in industrial processes and notably for the restoration of marginal lands and avenues of future research on this high-value multi-use plant for the global fiber market are described.

## 1. Introduction

*Urtica dioica* L. is named “the great stinging nettle” but is known colloquially and in literature only as “stinging nettle” [1,2,3,4]. The stinging nettle *Urtica dioica* L., with the small nettle *Urtica urens*, represent the most common species of the genus *Urtica*, which comprises 63 species of flowering plants and belongs to the family Urticaceae (40 genera and more than 500 species) of the major Angiosperms group. These two species are native to Europe, Africa, Asia, and North America [5]. *Urtica dioica* L. was described for the first time by Carl von Linné in 1753. The name “Nettle” could originate from the Anglo-Saxon word “noedl” which means needle, and *Urtica* is a Latin word that means “to burn”, which refers to the burning provoked when touching the plant. “*Dioica*” refers to the fact that male and female flowers are located on separate plants [3].

The stinging nettle (*Urtica dioica*) has been used for food and fibers at least since medieval times. Along with flax and hemp, nettle was the most important plant-based textile material in Europe because it grows even in northern climates, unlike cotton. Germany and Austria were pioneers in cultivating nettles during the 19th century and began the commercial farming of nettle fiber (Figure 1). With sanctions imposed on cotton during the First World War, the German army used nettle fabric for their soldiers’ uniforms. However, cheaper fibers from annual crops were more easily available after the Second World War [1]. More recently, growing concerns about the use of non-renewable resources in manufacturing have led to renewed interests in the reuse of natural or biobased fibers [6] (Figure 1). Forecasting studies predict further strong market development for fibers derived from plants, with an estimated increase of 300% over the next 25 years [7]. It has been predicted that land area needed for plant fibers for material uses could reach as much as 300,000 ha by 2035 [8]. In Europe, the main plants used are mainly flax and hemp, with a relative market share of the biofiber market of 64% and 10%, respectively [7]. However, there are good economic and ecological reasons for also growing *Urtica dioica* as a fiber crop: (i) it is a perennial crop with low fertilizer and pesticide requirements [9], (ii) there is a high cultivation potential in several areas that enable regional production [10], (iii) it may improve soils overloaded with nitrates and phosphates, as the nettle is a nitrophilous herbaceous plant [9], (iv) it promotes local flora and fauna diversity [10], and (v) it can be produced on land unsuitable for food production, including contaminated lands. Producing fiber crops on marginal lands which are unsuitable for food production might help mitigate potential conflicts between food and non-food production. The stinging nettle has a further card to play in this context, as it grows vigorously everywhere, without intensive inputs such as pesticides, herbicides, or irrigation, even in fairly poor soil [11]. Nettle also grows in cool climates [12], making it a relevant plant candidate for local production and processing across Europe. *Urtica dioica* is also often co-associated with poplars [11] and willows in riparian habitats across Europe [13], which are also intensely used in phytomanagement practices.

This review provides an overarching treatment of the emerging opportunity from nettle fiber production, in particular related to the repurposing of marginal lands and the production of renewable fiber resources. It is divided into four sections illustrated in Figure 1, describing (i) the history of nettle usages (ii) the biology, physiology, and genetics of *Urtica dioica* L., (iii) the cultivation, harvest, and fiber processing, and (iv) the various possible value chains and uses of nettle. It aims to summarize available knowledge on the use of nettle for fiber to provide a basis of understanding for its deployment, and also highlight where improvements in crop production and industrial fiber development would be beneficial. We identified new and rising fields in this research area, hopefully attracting new researchers.

## 2. Historical Perspective

Nettle has been part of human society for centuries and is widely valued for its seemingly endless array of uses. As nettle is coming under the spotlight again, we review the history of its selection, harvest, and fiber processing in this section.

### 2.1. Clone Selection

Stinging nettle (*Urtica dioica* L.) has been processed into textiles for hundreds of years. A preserved nettle tissue proves its use in Switzerland as early as the 7th century [14]. A first written link between “nettlecloth” and “Urtica” can be dated back to 1391 in Great Britain [15].

The first industrial attempts to use the nettle stem for supplying textile fibers date back to 1850 and the following decades [16,17]. Thus, it has a long history as a fiber plant in Germany and Austria where it was used, along with flax (*Linum usitatissimim* L.) or hemp (*Cannabis sativa* L.) for textiles before the appearance of cotton (genus *Gossypium*) [18]. Many nettle clones currently being developed for fiber applications derive from collections initially curated during this period. Gustav Bredemann, a German agricultural scientist and botanist, and others began in the 1920s with a collection of originally wild nettle plants and the selection of promising specimens regarding their vigor and fiber content [17]. Fiber nettles from this selection were cultivated and showed a fiber content up to 17.6% (pure fiber content after chemical separation), at least three times higher than that of wild nettles [17]. After 1945 interest diminished, although about 30 cultivars, respectively clones (Hamburger nettle assortment), were maintained and preserved by the Institute for Applied Botany of the University of Hamburg [19]. In 1991, the Agricultural Institute of the State of Thüringen” (TLL—Thüringer Landesanstalt für Landwirtschaft), began investigations of an additional set of nettle clones that the former Federal Research Centre for Agriculture had maintained (FAL, Braunschweig, Germany) [19]. Biomass yields of up to 90 dt ha^−1^ and fiber contents (pure fiber content) after chemical separation of up to 14% were found over a four-year period, based on a 1942 assessment method [20].

These clones were re-evaluated in 1993. Initially, work focused on investigating plant development, yield, and phenotyping of plant material because much of the original descriptions and performance data had been lost over time [21]. In 1993, Dreyer began with this range of plant material in a 4-year targeted multi-factorial cultivation experiment (clone, fertilization) with, besides others, the determination of phenotypical variability, characteristics of leaves and stem such as biomass yield and fiber content. Of the clones tested, “clone B13” was found to provide the best yields of fiber and stem dry mass [17] and was the most used in experimental field trials [2,21]. Francken-Welz reports on field experiments on sandy loam near Bonn (west of Germany) in 1997. The nettle basis is the Thuringian assortment, which does not allow any clear reference to the original Bredemann clone basis. Results of the investigated yield have been determined from 2 and 3 years of cropping. A biomass yield from 65 to 82 dt ha^−1^ with a mean fiber content of 17.3% was reported at planting densities varying from 1.7 to 5 plants m^−2^ and a mean nitrogen fertilization rate of 140 kg ha^−1^ [22]. It can be assumed that the fiber content was determined by laboratory-based, mechanical decortication. An additional result from the experiments is the documentation of a possible seed-based establishment of nettle. This method results in increased effort and risk, as well as lower competitiveness at the beginning of crop establishment. These results are supported by the experiment from the Thuringian work [23].

A research project focused on breeding was funded by the German Federal Environmental Foundation (DBU—Deutsche Bundesstiftung Umwelt, Osnabrück, Germany) from 2008 to 2012. The main goal was to develop a cost-saving multiplication procedure for nettles based on somatic embryogenesis and the creation of encapsulated synthetic seeds [24]. Aside from the generation of numerous fundamental knowledge of the process steps, the project was not successful, mainly due to phytosanitary reasons in the course of the generation and development of a callus. Furthermore, particular effort was put into breeding new clones with higher fiber content based on the original Hamburger assortment of Bredemann in a simultaneous project funded by the Federal Ministry of Food and Agriculture, Berlin, Germany. Three years and two locations (Hannover and Soltau, Lower Saxony, Germany) trials with selected Bredemann and new bred clones showed a high variability of phenotypical and yield-related characteristics. In particular, breeding is, therefore, more difficult because reliable selection requires a corresponding database from many years of cultivation in the field [25]. The work’s starting point was six original clones of Bredemann and four others already bred (“Z”) from them. A selection of eight new cultures (of which five have Bredemann’s clone B 13 in parents’ generation) was characterized as promising for further multiplication and testing based on a multi-criteria assessment including agricultural, yield, and quality aspects. As an example, a maximum fiber content of 16.7%, as determined with laboratory equipment, could be obtained for one of the new genotypes within the first two-field test years. Last but not least, a research project is mentioned in which breeding and agricultural aspects, such as processing aspects in a value chain, were investigated from 2015 to 2018 [26].

### 2.2. Fiber Processing

Figure 2 illustrates the development of nettle fiber extraction technology from the 18th century to the present. Three hundred years ago, a standard process line to produce nettle fibers for manual spinning included the following process steps: manual harvest of stems, sun drying, breaking the defoliated stems, rinsing and drying, and finally hackling them (line; [27] cited in [28]. Another manual process line for nettle textiles that is still practiced today in western Nepal is as follows ([29]; not shown in Figure 2): after manually harvesting the fiber plant (*Girardinia diversifolia*—Urticaceae; “Allo fiber” according to [30]) with an iron sickle, the leaves and stinging hairs are rubbed off, and the bark is stripped off manually directly on-site after manual breaking. For the separation (degumming), the ash-covered bark is cooked in water for four hours, followed by repetitively beating the bast with a wooden hammer and water rinsing. Afterward, the fiber bundles are refined by placing the bast in a clay/water mixture and, after sun drying, beating the clay-covered bundles again with the wooden hammer. The stretched and parallelized fiber bundles can then be used for manual spinning [29,31].

Before the First World War, the “traditional” process line to produce nettle fibers for spinning was very similar to hemp fiber production [32]. After a manual harvest of nettle plants, the stems were traditionally either water or field retted [9,32]. Retting is the process employing the action of micro-organisms and moisture on plants to dissolve or rot away much of the cellular tissues and pectins. After drying and peeling off the leaves, the decortication was done using a breaking unit, the bast fiber bundles were separated via scutching to separate the impurities from the raw material, and the refining was performed via cooking and hackling (second line; [33]). von Roeßler-Ladé recommended cooking nettle fibers before hackling [32]. That differs from processing hemp, which was cooked only after spinning. Ganswindt reports about the increase of nettle fiber production for Germany’s textile industry during the First World War (1914–1918) because the stocks of flax and cotton had been consumed [34]. School kids were collecting (“harvesting”) nettle stems and got 6 to 14 D-Mark (3–7 €) for 100 kg of stems, depending on the length [34]. After drying and removing the leaves, chemical retting with an ammonia solution was performed to degrade the pectin lamella. Alternatively, water retting was conducted. Afterward, the stems were decorticated using a breaking unit, and further, the fiber bundles were hackled to splits, and the fibers were straightened. After hackling, the isolated but rough nettle fiber bundles were cooked in a soap solution to achieve a nettle fiber quality that could be spun on cotton spinning machinery. During several trials in his spinning mill, the director [35] found out that the isolated single nettle fibers were spinnable. This process line is described by [34] and is shown in Figure 2 in the third line.

During the Second World War (1939–1945), the research on using nettle fibers in the German textile industry increased again. Bredemann described two main methods that were developed during this time: the flock bast procedure (shown in Figure 2, line 4) and the so-called Elster procedure (named after the textile entrepreneur Johannes Elster of the company Gebrüder Uebel in Germany) [16]. The nettle stems were not retted for the flock bast procedure, just dried to a specific condition (7–9% water content), defoliated, and subsequently decorticated using a breaking unit [36]. Afterwards, the bast was cleaned using a comb shaker, cooked in a sodium hydroxide (NaOH) solution, and carded before spinning [16]. In contrast, the Elster procedure includes different cooking processes in water with adequate substances. For the whole processing, a machine was employed, which is protected by several patents [37,38]. Unfortunately, this machine was destroyed during the last war days [16].

A sustainable alternative for the chemical processing methods was presented by Dreyer et al., using a degumming process with enzymes to remove phosphatides from crude oils to improve physical stability and facilitate further processing and achieve nettle fibers with textile quality [39]. Dreyer et al. [39] used field retted nettle stems in their research. A disordered fiber line (total fiber line) using a purely mechanical process (known for hemp; see [40]) was used for the decortication and separation process, using a breaking unit and a coarse separator. The process line described by Dreyer et al. [39] is shown in Figure 2, line 5. Enzymatic processing has environmental advantages compared to chemical processing [39]. Based on the work of Dreyer et al. [39], the production on a roughly 50 kg scale for the enzymatic treatment of bast fibers was implemented semi-industrially [41]. An industrial process line, which is nowadays used for the production of textile grade fibers by the company NFC GmbH Nettle Fibre Company (Dahlenburg, Germany), is described in [26] and is shown in Figure 2, line 6. Cultivated nettle clones are mechanically harvested, field retted, and dried. A hammermill is used for decortication. The cleaning is divided into two steps: a tambour and a step cleaner. Finally, the fiber bundles get refined using an opener [26]. For spinning fine yarns, degumming and carding are performed. Neglecting the refining steps of enzymatic or chemical degumming and carding (necessary for spinning single nettle fibers or fine fiber bundles into fine yarns; shown in Figure 2, lines 5 and 6), coarse fiber bundles for technical applications like needle felts can be produced.

For the first time, the possibilities of agricultural production of nettle with comparatively large-scale cultivation, and quasi-industrial fiber extraction, could be integrated within the framework of a research project [26]. Established (e.g., clone B 13, “Z” clones), and for the very first time, new “L” genotypes were grown under different fertilization and plant density schemes to assess yield, processing, and quality aspects. A further experiment on fiber processing demonstrated that values of fiber content found in the literature are highly dependent on the extraction methods (e.g., chemical determined (“pure”) fiber content versus mechanical processing with laboratory decorticators).

## 3. Biology, Physiology, and Genetic of *Urtica dioica* L.

The objective of the following section is to describe the current knowledge on the biology, genetic, and ecology of nettle while focusing on stem anatomy and the morphology and composition of the bast fibers. This knowledge is crucial to better apprehend the various steps in nettle selection, straw processing, and fiber extraction, and to evaluate the fiber yield and analyze fiber features and suitability for textile and material applications.

### 3.1. Biology, Ecology, and Reproduction

*Urtica dioica* is found in many cold to temperate regions of the world: Africa, America, Asia, Australia, and Europe. The stinging nettle is widespread in Northern Europe and Asia, and less widespread in Southern Europe and Northern Africa [42]. It is also widely distributed in North America, especially Canada and the United States, and is growing in abundance in the Pacific Northwest, especially where annual rainfall is high. According to Darwin’s transoceanic diffusion hypothesis, nettle seeds are viable after having floated in seawater over the long term, favoring long-distance diffusion, which may explain its wide geographical distribution [43]. At the subspecies or variety level, distributions are very different. For example, *Urtica dioica* subsp. *gracilis* appears to have a geographical distribution limited to America, while *Urtica dioica* subsp. *dioica* is widely distributed [42].

The stinging nettle is a herbaceous nitrophilous perennial plant that grows in a wide range of habitats, as a common species of riparian habitats, swamps, meadows, riverbanks, wastelands, floodplains, and disturbed areas. Large monospecific stands have been reported on marginal sites (e.g., slag heaps, yards) rich in nutrients [44]. It is also frequently found under near-shore willows throughout Europe [45]. It prefers moist, rich soils and can thrive in full light but does best in semi-shade. According to Taylor [12], *Urtica dioica* subsp. *dioica* cannot support anoxic conditions for long periods, for example, as might occur with flooding [46]. Nettles prefer loose soils with organic matter and high nitrogen levels for rapid growth [47]. Nitrogen stimulates the growth of aerial parts, and it has been suggested that nitrogen is the most important component of nettle nutrition [48].

The species is morphologically quite plastic [49] and thus encompasses a large number of subspecies on all continents [42]. However, all subspecies are erect with yellowish and cylindrical rhizomes and stolons (Figure 3a). The root system is found in the organic horizon at shallow depths (10 to 30 first cm). The stem has a quadrangular section (Figure 3b) and could reach up to two meters high with leaves in opposite pairs (Figure 3c), oval or even lanceolate, with a rounded or cordate base, toothed leaf margins, and an acute or acuminate leaf apex (Figure 3d). Two opposite leaves and four stem stipules are inserted at each node, except for the cotyledonary node and the first node of the main stem, which have no stipules [12] (Figure 3e). Young leaves are the organs of nettle that contain the most moisture (twice as much as in the roots), making them particularly tender [50]. The inflorescences are axillary, spiked, and four per node, with many small green unisexual flowers [51]. Flowers appear from June to October [47], with male and female flowers usually found in different plants (e.g., *Urtica dioica* subsp. *dioica*). However, dioecious (*Urtica dioica* subsp. *dioica*) and monecious (*Urtica dioica* subsp. *gracilis*) subspecies have been reported [52]. Große-Veldmann and Weigend [53] also suggested that no strict dioecious species belong to the genus *Urtica*. *Urtica dioica* and all its subspecies are polygamous, with *Urtica dioica* subsp. *dioica* mainly represented by 80 to 90% of dioecious individuals and 10% of monoecious individuals (basal male inflorescences and apical female inflorescences). Male flowers are more erect, yellowish, with four long-filleted stamens folded into the flower bud, whereas female flowers are greenish, with a unilocular ovary topped by a brush style and stigma, and tend to be more hanging [54,55] (Figure 3f,g). Fruits are achenes, light (0.2 mg), very small (1.3 × 1.0 mm^2^), and therefore easily carried by wind [56]. The woody stem accounts for 23–30% of the total biomass [57]. However, abiotic factors such as altitude seemed to induce morphological and anatomical variations in stinging nettle populations [58].

Stinging nettles are covered with hairs on the leaves (Figure 3h) and on the stems (Figure 3i), called trichomes, represented by short simple hairs and longer rigid hairs that sting. Trichomes density is lower at the base of the stem, at the internodes, and on the upper surface of the leaves [59]. When they break, the small tubes release a liquid containing formic acid (methanoic acid; CH_2_O_2_), serotonin (5-hydroxytryptamine; C_10_H_12_N_2_O), histamine (2-(1H-imidazol-4-yl)ethanamine; C_5_H_9_N_3_), and acetylcholine (2-acetoxy-N,N,N-trimethylethanaminium; C_7_H_16_NO_2_) [60,61]. These compounds cause itching and burning and thus serve as a defense mechanism against insects, herbivorous mammals, or heavy grazers [5]. This mechanism is both mechanical and biochemical [62]. There is evidence that trichomes have evolved to defend nettle against herbivorous mammals [63]. Indeed, populations under intense grazing by mammals have more trichomes than populations in areas with less intense grazing [64]. Phenotypic plasticity is an important trait for *Urtica dioica*; for instance, fewer trichomes were produced when grown in the shade rather than in the sun in a culture experiment [65]. These experiments have also hypothesized that these variations (including polymorphism in hair density) appear to be genetic and hereditary [66].

The main strategy for the development of nettles is underground vegetative propagation. In late summer, when leaf dehiscence occurs, stems lodge to form a rhizome. A new individual can be produced from each stem node or older rhizomes [44]. Shoots from rhizomes develop mainly in the fall, over winter, and resume growth the following spring [67], although some may die. About one-third of the maximum shoot biomass is maintained during the winter [44]. Dreyer [17] concluded from his research a distinct earlier shoot growth in late winter/spring based on a comparatively large number of shoots as well as high nutrient storage in its rhizome. This reproductive strategy explains the invasive character of nettle, forming dense monospecific or even monoclonal stands in environments where other plants have low competitive ability or when soil conditions are favorable. Thus, root length measurements have shown that nettle is able to develop more than a third more compared to, e.g., barley, oat, or beans in the 25 cm top layer of the considered layer [17]. Stinging nettle can also reproduce sexually. However, sexual reproduction has little impact on its spread, but it has been described to be essential for the colonization of new sites [68]. One single stinging nettle plant can produce up to 20,000 seeds in open areas, while this number is reduced to 5000 in shaded areas [69]. Seeds are usually sown from August [70], and nettle seeds can remain viable for a long time in salt water (until 240 days) [43] and survive ingestion by animals [12]. Dissemination of nettle seeds is not based on a single predominant strategy but on several mechanisms (e.g., wind, water, insects) that are more or less effective depending on the plant’s environment [12,70]. In temperate climates, germination usually begins in early January and peaks in April [12]. Germination is inhibited by darkness and stimulated by light and temperature fluctuations [71], so sexual reproduction is not very efficient in sites with high vegetation cover.

### 3.2. Stem and Fiber Morphology and Fiber Composition

In the open literature, the most detailed description of the anatomy of the nettle stem and, more broadly, of the vegetative organs of the nettle is undoubtedly in the memoir by Auguste Gravis published in 1885 [72] as depicted in Appendix A. It is based on the examination of a great number of stems, leaves, and roots, including no less than 15,000 sections (transversal but also radial and tangential) made at different degrees of development, at different ages, and considering growing conditions. If this investigation was originally intended to serve as a basis for botanic classification, it constitutes today one of the most detailed studies on the anatomy of the nettle.

In recent years, the bast fibers have been the subject of more characterization due to their potential application in textiles or other materials. For example, analyses were made on wild nettle in the frame of the PHYTOFIBER project (www.phytofiber.fr (accessed on 1 September 2018), [11]). Figure 4 shows several examples of the typical transverse cross-sections of the stems. The diameter of primary bast fibers ranged from a few microns to a maximum of 100 µm, with a wall thickness from one µm to more than 20 µm, depending on the age, stem position, and plant maturity. In agreement with the observations of Gravis [72], it can be seen (Figure 4b) that the primary bast fibers are sometimes collapsed, depending on the cell wall thickness when the diametric growth begins in the stem. No secondary bast fibers were observed in the analyzed stems. Bacci et al. [2] reported, for the bast fibers of cultivated nettle (German fiber nettle clone 13), mean diameters of 19, 32, and 47 µm in the top, middle, and bottom parts of the stem with an average length of 58, 50, and 43 mm in these same parts. Regarding their biochemical composition, Dreyer and Edom [73] values compiled from literature with approximately 54% of cellulose, 10% of hemicelluloses (mainly composed of arabinan, xylan, galacturonan), 4.1% of pectins, 9.4% of lignin, 4.2% of wax and fats, and 18% of water-soluble products. This chemical composition is highly affected by retting or the different methods used to extract the fibers [9,74] and harvest date [2]. After retting, the cellulose content can reach values up to 88%, while the other constituents’ content drops to 4%, 0.6%, 5.4%, 3.1%, and 2.1%, respectively.

### 3.3. Phylogeny and Genetic Features

The genus *Urtica* is very characteristic and easy to identify, but species delimitation is still problematic, especially for *Urtica dioica* L., which has more than 20 infraspecific taxa recognized in Eurasia and America [75,76]. Große-Veldmann and Weigend [77] have identified five morphotypes of *Urtica dioica* subsp. *dioica*: var. *dioica*, var. *hispida*, var. *sarmatica*, var. *Holosericea*, and var. *glabrata*. These morphotypes are distinguished in terms of habitat preference, geographical distribution in Europe, variation of indumentum (e.g., trichome density and number of stinging hairs), leaf shape, and leaf edge morphology (e.g., slightly or largely oval, cordate or truncated base) [42,78].

The level of ploidy is a second reason justifying the recognition of infraspecific taxa. Polyploidy shapes the pattern confining diploid cytotypes to residual habitats [79]. In the two most recent phylogenies [42,80], *Urtica dioica s.l.* forms a well-supported clade consisting of different related taxa, whatever the markers used (nuclear, chloroplastic, or a combination of these two types). This clade falls into a western Eurasian clade and an Asian–American clade. More precisely, the western Eurasian group includes all the *U. dioica s.str*. and shows a sister relationship with the endemic Mediterranean group (i.e., *U. atrovirens*, *U. bianorii*, and *dioica* ssp. Cypria) and the two related African *U. massaica* and *U. simensis*. The second clade groups the western and north American *U. gracilis* in a subclade sister to the Asian–Australian subclade. However, the different subspecies, or even varieties, present morphological differences that could be due to phenotypic plasticity rather than genetic divergence [80]. Rejlová et al. [79] thus suggest that genome size can contribute to the delimitation and detection of closely related species (e.g., *Urtica bianorii* and *Urtica kioviensis* showed larger genome sizes in their study). Differences in genome size values may indicate genetic distance. However, the genome size does not delimit the subspecies, and only the level of ploidy is accepted as a delimiting trait of *Urtica dioica* subsp. *dioica*. Phylogenetic data obtained using molecular markers show that the morphological and geographical characteristics used to distinguish and group species do not reflect phylogenetic relatedness [81]. Standard molecular markers do not resolve relationships at the subspecies or variety level. Große-Veldmann [80] has therefore used a genotyping-by-sequencing (GBS) approach on 53 taxa. They used the cutting enzyme PstI-HF (recognition site: CTGCA’G) and the methylation-sensitive enzyme MspI (recognition site: C’CGG) was also used to understand evolutionary relationships within a complex of species belonging to the genus Cycnoches, a tropical orchid [82], and to deduce the phylogeny of seven closely related species of the genus *Carex* [83]. They obtained 4013 loci and 30,840 SNPs. However, this did not allow the identification of the infraspecific relationships of *Urtica dioica sensus stricto.* Separate treatment of the different alleles did not improve the resolution either. Previous results from studies of phylogenetic relationships based on standard nuclear markers (e.g., ITS, trnS-trnG, trnL-trnF, psbA-trnH) have given mostly the same results. Farag et al. [84] showed that there is little similarity between 43 secondary metabolites groups (mainly phenolic compounds and hydroxyl fatty acids) and phylogenetic data. However, one subgroup is recovered in both analyses: *Urtica dioica*, which appears as an exclusive group.

The majority of published estimates of plant genome size have been made using flow cytometry [85]. This method consists of estimating the DNA content of isolated nuclei stained with a DNA-selective fluorochrome [86]. DNA content of a haploid cell is usually measured by the C-value expressed in picogram (pg), with 1 pg equals to about 978 Mb [87]. For *Urtica dioica s. str.*, estimates of genome size range from 597 to 1540 Mbp (Table 1) [80].

### 3.4. Nettle Phytochemistry

Various metabolomic approaches (e.g., gas chromatography-mass spectrometry (GC-MS), ultra-high-performance liquid chromatography-high-resolution mass spectrometry (UHPLC-HRMS/MS)) have been applied for unraveling the content of metabolites in *Urtica dioica*. Various compounds that may have nutritional and/or medical importance were detected with these screening approaches. Shokrzadeh et al. [95] showed that *Urtica dioica* extracts have therapeutic potential for the attenuation of oxidative stress and diabetes-induced hyperglycemia. A large variety of compounds might be responsible for these effects. Indeed, a huge diversity of secondary metabolites was detected by Al-Tameme et al. [96], who reported aromatic rings, alkenes, aliphatic fluoro, alcohols, ethers, carboxylic acids, esters, nitro compounds, hydrogen-bonded alcohols, and phenols in methanolic extracts of *Urtica dioica*. Likewise, alkaloids, saponins, tannins, flavonoids, steroids and terpenoids, polyphenols and cardiac glycosides were detected in *Urtica dioica* leaves based on Fourier transform infra-red (FTIR) spectroscopy [97]. Pinelli et al. [98] reported that chlorogenic and 2-O-caffeoylmalic acid dominated the phenolic compounds in leaves, whereas in stalks, mainly flavonoids and anthocyanins were found. More specifically, Grauso et al. [99] found that stinging nettle extracts contained two pentacyclic triterpenols α- and β-amyrin in the non-polar fraction, whereas, in the polar extract, large amounts of choline were found. Brahmi-Chendouh et al. [100] reported that hydroxycinnamic acid derivatives next to C-glycosylated flavones were the most representative constituents in stinging nettle leaves. However, taxa belonging to *Urtica dioica* appear to have similar metabolite compositions and, therefore, similar pharmacological properties [84]. Modern hydroponic cultivation methods in greenhouses have also been set up for the easier management of environmental factors to improve metabolite production [101]. Nevertheless, males and females harbor different content and chemical composition of polyphenolic acids in their leaves, with the male form characterized by a higher content of these compounds [102]. Phenolic compounds in nettle leaves are strongly influenced by the habitat and other several biotic and abiotic factors [103,104]. Total phenol content is also different depending on the phenological stage of the nettle [103,105]. The highest polyphenol content seemed to occur between April and July at the beginning of the vegetation period [106,107]. More precisely, it decreased in leaves from spring to autumn while a slight increase was observed in roots [108]. A recent paper highlighted the fact that the biological resources of wild-growing types of *Urtica dioica* L. from the European south of Russia are a valuable source material for obtaining varieties with valuable biochemical characteristics [109]. Being a silicon-rich plant, the stinging nettle also represents a valuable interest for cosmetics and natural medicine [110]. On the contrary, other compounds such as allergic proteins were retrieved and can cause rhinitis in humans as shown using an allergomic approach; however, further research is needed to assess the allergenic potential of *Urtica dioica* [111].

Literature reports of biomass concentrations of Ca, Mg, and K in *Urtica dioica* are variable (Appendix A). In [112], K resulted in being the major bulk element in leaves (33.9 g kg^−1^), followed by Ca (28.6 g kg^−1^) and Mg (8.69 g kg^−1^), within the same order of magnitude as those published in a recent paper [113]. In Kara [114], Ca resulted in being the major macro-nutrient in nettle herbal infusion (seemingly leaves, 38.4 g Ca kg^−1^, 17.5 g K kg^−1^, 7.32 g Mg kg^−1^), while [115] reported higher Mg concentrations in leaves of *Urtica dioica*, in the range of 25.1–35.6 g kg^−1^. Comparing the elemental composition of stems and leaves, Mg and Ca resulted in being, respectively, 2 and 3-fold more abundant in leaves than in the stems of *Urtica dioica* [115]. Iron is reported to be the most important TE (trace element) in stinging nettle [112] and the major TE in leaves, with values ranging from 151 mg kg^−1^ [112] to 999 mg kg^−1^ [114]. Compared to TE concentrations in plant leaves [116], Cu, Zn, Cr, and Co concentrations in stinging nettle for leaves and whole plants [11,112,114,115] can be considered within physiological levels when growing in non-contaminated soils; exceptions are reported in [117], where Zn and Cr concentration in whole *Urtica dioica* plants exceeded physiological values in leaf crops [116]. Manganese concentrations in stinging nettle vary considerably according to different sampling locations. Specimens from Macedonia were found to be Mn deficient in all locations analyzed [115] as compared to physiological Mn concentrations in plants [116], while in samples from Belgium [117] and Serbia [112], Mn concentrations in nettle biomass were in the physiological range. Pb, As, and Hg in *Urtica dioica* are well below “toxic” plant levels [116] in all literature values, with an exception made for Pb in stinging nettle samples from uncontaminated soil in Belgium (34 mg kg^−1^, [117]).

### 3.5. Nettle-Associated Organisms

Little information about the organisms associated with nettle roots is available [118]. Toubal et al. [119] studied the bacterial diversity associated with different tissues of nettle using biochemical tests and spectrometric analyses (MALDI-TOF MS), resulting in the isolation of 7 genera and 11 species belonging to the genera *Bacillus*, *Escherichia*, *Pantoea*, *Enterobacter*, *Staphylococcus*, *Enterococcus*, and *Paenibacillus*. For the different tissues of nettle, the most common species identified was *Bacillus pumilus*. Nettle has also been shown to host *Ralstonia solanacearum*, which causes potato brown rot [120]. Mojicevic et al. [121] isolated *Streptomyces* spp. from the rhizospheric soil of *U. dioica*, which has the ability to produce antifungal compounds against *Candida krusei*, *C. parapsilosis*, and *C. glabrata*.

Microscopic observations of stained (Trypan blue) and labeled (WGA-AF488) root segments of *U. dioica* revealed that fungal structures colonized the cortical cells, which includes microsclerotia formed by dark septate endophytes and spores (Figure 5). Other observations revealed a relatively low rate of AMF structures in the roots [122]. The partial diversity of the root-associated fungal microbiome of nettle grown at the metal-enriched sediment disposal site of Fresnes-sur-Escaut was recently characterized using the Illumina MiSeq technology approach [123]. The nettle mycobiome was dominated by Pezizomycetes and Leotiomycetes, including endophytic and saprotrophic taxa (Figure 5), with the supposedly saprotrophic genus *Kotlabaea* being the most abundant. In terms of diversity and abundance, Pezizales, Helotiales, Pleosporales, Agaricales, Hypocreales, and Thelephorales were the most represented orders [123]. Only 54% of the fungal sequences were successfully assigned to a genus, reflecting the lack of data about the nettle microbiome. In addition to *Kotlabaea*, *Olpidium*, *Tetracladium,* and *Hymenoscyphus* were among the most abundant identified genera. Despite being a known non-AMF plant, nettle was associated with a significant proportion of ectomycorrhizal OTU (9.7%), suggesting some connections with the symbiotic mycobiome of surrounding poplars [123].

The first work on nettle insects was done in Europe in the 1970s [124,125,126] and highlighted its importance as a reservoir of insects, hosting a large diversity and particularly some species of Hemiptera and Coleoptera [127]. Nettle hosts more than 100 insect species, thirty of which are classified as specialists, including several species of Aphididae, Psyllidae, and Nymphalidae. Many other generalist species belonging to Miridae, Lygaeidae, or Cicadellidae families have been found, as well as a complete set of predators and parasitoïds such as Coccinellidae, Syrphidae, and Braconidae species. More recently, studies conducted in the United States [128] and in Belgium [129,130,131] confirmed that maintaining *U. dioica* as patches or more efficiently as a monospecific stand [132] contributed to the improvement of biodiversity within agrosystems. Nettle could specifically promote beneficial insects, including several taxa of predators (e.g., Coccinellidae, Syrphidae, Anthocoridae) and parasitoids (e.g., Braconidae, Diapriidae), known as natural competitors of crop pests [129,131,133]. The promotion of natural enemies of potential pest species is linked with the hosting of *Microlophium carnosum*, an aphid serving as a diversion or alternative prey for many predators present on nearby infested crops [134]. In addition, a plantation of a nettle clone selected for its fiber content was shown to be able to support this aphid species and thus promote its natural enemies [135]. Aphids hosted by nettle can be a food source that can attract aphidophagous or parasitoid insects [133]. To encourage the migration of nettle-related predators in the surrounding agrosystems, Alhmedi et al. [129] suggested harvesting nettle rapidly after the arrival of aphids. The stinging nettle could consequently contribute indirectly to the regulation of crop pests in the surrounding environment [131].

Succession is a process motivated by positive interactions between species, with pioneer species facilitating the colonization of an environment by less stress-tolerant but more competitive species [136]. Thus, facilitation represents the most important process in successional sequences, particularly in primary succession, where environmental conditions are difficult [137]. During primary succession, soil N is often the limiting factor in vegetation establishment [138]. *Urtica dioica* is described in the literature as a highly competitive species when moisture and soil conditions are favorable, in particular thanks to its ability to multiply vegetatively efficiently with stolons and rhizomes. Compared to other plants (e.g., *Agropyron repens*, *Artemisia vulgaris*, *Calamagrostis epigeios*, *Cirsium arvense*, *Epilobium angustifolium*, *Phalaris arundinaceae*, *Typha* spp.), the stinging nettle showed the greatest capacity for expansion [139]. However, when soil conditions are not optimal, many herbs such as thistle, teasel, *Taraxacum officinale* [140], or *Gallium aparine* [141] can inhibit its development. Additionally, a study on the cultivation of *Urtica dioica* L. with other plant species highlighted that nettle was well established in a soil that was originally sown with leguminous species, with a significant proportion of wild white clover. However, its ability to become established is considerably reduced in environments dominated by some grasses [142]. This indicates that pre-planting tillage is an important factor in the development of this plant, even though *Urtica dioica* is considered a weed in intensive agriculture [9]. Several studies sought to demonstrate the allelopathic potential of nettle, which depends both on the concentration used and the plant species. For example, some studies showed the inhibitory effect of aqueous nettle extracts on barley [143], wind grass, and lambsquarters germination [144], whereas [145] showed that leaf and root extracts of *Urtica dioica* significantly reduced root mass. Khan et al. [146] found a strong inhibitory effect of methanolic extracts of the stinging nettle on radish germination and growth.

Recent genetic studies demonstrate that *Urtica dioica* undergoes a huge gene flow resulting in a high recombination rate and that morphological differences can be traced back to local directional selection and phenotypic plasticity. Molecular approaches based on environmental metabarcoding further allow deciphering the fungal microbiome of nettle, which was dominated by Pezizomycetes and Leotiomycetes. This approach has extended to a large variety of sites to characterize the core microbiome of nettle better.

## 4. Nettle Cultivation, Harvest, and Fiber Processing

This section covers the topic of the production, extraction, and processing of nettle fiber. The following essential questions are to be answered, focusing on the overall idea of the publication, i.e., evaluating the possibilities of using nettle as a resource for biobased materials:Can the cultivation of nettle be improved to increase bast fiber yield?Are there processes on the market for processing bast fibers that can be used for nettle processing without major adaptations?Which processes can be chosen to obtain nettle fibers for high-quality fiber composites?What still needs to be optimized to turn nettle into a valuable fiber for composites?

### 4.1. Agronomic Practices for Fiber from Nettles

None of the nettle clones developed for fiber applications tested for cultivation [21] have been officially registered (see Section 2.1 for details on early selection programs in Germany). Nettle “clones” designate genetically identical nettle plants obtained by vegetative propagation. While nettle can be propagated by seeds (see Section 3.1), the young plants compete poorly with weeds [140] and the parents’ heterozygosity level is high, resulting in non-homogeneous plants [9,21]. Additionally, fiber content may be lower when plants are grown from seed [2]. Vegetative propagation can thus be used [9,140], with top in vitro cuttings grown in a greenhouse [10,16,21]. Propagation medium and optimal season for harvesting mother plants are reported in Gatti et al. [147] and Di Virgilio et al. [9]. Direct fielding of cuttings is possible [148,149]. When cultivated by rhizome cuttings, it appears that the morphological differences and developmental characteristics are different between male and female plants. Indeed, male plants would produce more leaves, more side shoots, and more rhizomes [150]. None of these possible cultivation techniques is well described, and further research is needed to identify the most suitable technique depending on the objective of the cultivation [101].

Soil preparation is similar to that of other fiber crops. Depending on the climate, an Autumn/winter plowing followed by a fine seedbed in the spring with power harrowing is advised for successful transplanting. Cuttings, when well rooted, are transplanted with conventional crop (e.g., tomato, cabbage) planting machinery in early autumn in moderate climates or late spring in a cold climate. Different planting densities (from 1.7 to more than 5 plants m^−2^) have been tested for fiber and medical applications (Table 2), with an interrow varying from 50 cm to 1 m.

The analysis of literature data indicates that increasing planting density results in a significant decrease (*p:* 0.013) in stem yield but not in fiber yield (Figure 6). The optimal planting density for fiber nettle ranged from 2 to 3 plants m^−2^ (Table 2). Bacci et al. [2] reported that a planting layout of 50 cm × 50 cm or 50 cm × 75 cm is optimal for high fiber and stem yield. No information is available on the effect of the preceding crop on nettle growth and productivity. Termination of nettle plantations requires deep fall tillage combined with spring grubbing and harrowing [16] or rhizomes grinding with forestry mulcher in spring. The duration of nettle plantations is given as a maximum of 4–5 years [16,151]. Plantations can last longer if weeds are controlled; 10 to 15 years, according to Vogl and Hartl [10]; and six years, with production years ranging from the second to the sixth year, with a maximum in the third and fourth years, according to Tavano et al. [159].

Fiber nettle is well adapted to various environmental conditions and has been successfully grown in most European climatic zones (Table 2). Different authors reported the need for rainy spring and summer after transplanting or several irrigation events until the crops are well established [2,9,10]. In general, fiber nettle, as a nitrophilous perennial plant that grows on ruderal sites, requires moist soils rich in organic matter, nitrogen [1,2,140], and phosphates [47]. Although no data on water use efficiency are available, fiber nettle is considered a water-demanding crop [2,9,10,21]. Nettle does not tolerate long flooding [16], and Šrutek [160] noticed that the presence of a phreatic table close to the soil surface limited the biomass production of nettles and reduced the length of their stems. The optimal soil pH range is 5.6–7.6 [9]. To date, no pesticides have been registered for nettle cultivation. However, weed management is essential in the first year after establishment [106]. Row spacings wider than 50 cm allow mechanical weeding with conventional maize weeder [161]. False seedbed sowing followed by selective herbicides against annual grassed and broad-leafed summer weeds can help control weeds in the first months after transplanting [162].

Stinging nettle is a perennial crop with low input requirements. The effect of N fertilization on fiber nettle yield and quality has been addressed in eight studies [1,2,4,21,151,155,156,158] (Table 2). The results of the linear regression analysis of nettle yield response to N fertilization (Figure 6). Figure 6 showed that stem yield increased significantly (*p*: 0.032) with the increase in N doses (43 kg DM per kg of N), while fiber content showed a slight but not significant (*p*: 0.228) increase (6.4 kg DM per kg of N). From the analysis of published studies, before transplanting, the addition of 70–80 kg N ha^−1^, 40–50 kg P_2_O_5_ ha^−1^, and 130–150 kg K_2_O ha^−1^ can be suggested in soils with medium to low fertility levels. Different low input cultivation systems have been tested for fiber nettle. Intercropping with fast-growing legumes species (*Trifolium* or *Vicia* spp.) [21,156] or moderate organic amendments (slurry, manure) application (60 kg N ha^−1^) were suitable to achieve consistent stem yields (>3 Mg DM ha^−1^) [1].

Nitrogen in nettle is mainly contained in free amino acids, with asparagine and arginine accounting for up to 80% and primarily stored in roots and rhizomes [163]. Nitrogen deficiency leads to reduced growth, reduced leaf area, reduced plant dry mass, and increased root dry mass [12]. The form of the fertilizers used will influence the response of the plant. Indeed, if ammonium is provided as the only nitrogen source, plants do not survive [164]. Additionally, ammonium nitrogen is far less beneficial than nitrate to the nettle. The chemical composition of stinging nettle can be affected by the age of the plantation, harvest time [105], or by nitrogen fertilization [165,166]. Nitrate content in leaves and stems increases when the nitrogen fertilizer dose increases, with stems containing up to 9 times more nitrates than leaves. Nitrogen fertilizer application also increases plant development, yield, and dry matter production of leaves and stems [167]. Nitrate accumulation in plants is affected by various factors such as genetic, environmental (e.g., humidity, temperature, photoperiod), and agricultural (e.g., nitrogen doses, nutrient availability) factors [168]. Increasing the dose of fertilizer leads to an increase in the aboveground biomass of the nettle and, conversely, would decrease the underground biomass. Thus, nitrogen is not the factor limiting root growth [169]. No significant pests and diseases for stinging nettle are known. Some fungi, aphid, and caterpillar attacks have been reported in the literature on young plants [9,10,16,21]. Four biostimulants that can be useful under stress conditions were tested on a nettle crop without effect on the yield or the morphology of the plants, but the content of specialized metabolites and minerals was impacted [170].

Harvesting nettles for fiber is typically carried out from the second year after establishment, at seed maturity between early August and mid-September [10,151]. No effects of delayed harvesting were observed on fiber quality, while the opposite was observed for premature harvest in summer [9]. Multiple cuttings can be performed if the crop is harvested for multipurpose destinations (e.g., leaves for medicine applications and cosmetics) [9]. However, no information on the fiber quality of stems re-grown after the first cuts is available.

### 4.2. Fiber Yield Improvement

It was reported that fiber content seems to be mainly influenced by genotype and little by cultivation method and environmental conditions (e.g., year, plant spacing, under seeding) [10]. These results contradicted those of Bredemann, who concluded on his experimental background at different locations that weather conditions (especially drought that has a negative effect) and nutrient supply could impact the fiber content of nettle. When nettle is cropped, the plants cannot be harvested in the first year for fiber production and must be cut to stimulate them to produce more aboveground parts. The maximum fiber content of 10% for cultivated nettle and 5% for wild nettle was reported in the first year of cultivation [2]. The pure fiber content differs according to the part of the stem considered, with a high content in the central part of the stem. For example, after 169 days of cultivation, 9.9% fiber was reported in the lower part of the stem, with 13.4% and 6.5% in the central and upper parts, respectively [2]. At the same time, the upper part can be used for valuable purposes such as medicinal, cosmetic, and food applications [4].

A literature data search was conducted to evaluate the stem and fiber yield potential of spontaneous and cultivated nettle as affected by age. English peer-reviewed and German publications reporting yield data from field experiments were searched using online search engines (Google Scholar, Scopus). From each study, the mean values of the stem and bast fiber yield (Mg DM ha^−1^) were extracted, together with the year of cultivation, N fertilization rate (kg N ha^−1^), and planting density (plants m^−2^). The nettle yield database is composed of fourteen publications resulting in *n* = 33 and *n* = 20 stem and bast fiber yield data, respectively (Table 2) [1,2,4,11,21,140,151,152,153,154,155,156,157,158]. The analysis of available data showed an average yield potential for cultivated nettle, from the 2nd year after planting, of 6.33 Mg DM ha^−1^ and 0.79 Mg DM ha^−1^, respectively, for stem and bast fiber (Figure 7).

Cultivated nettle showed the following yield progression for years 1–4 (median values): stem yield (3.86, 4.97, 6.15, and 6.18 Mg DM ha^−1^) and bast fiber (0.16, 0.45, and 0.89 Mg DM ha^−1^) (Figure 7). No data are available for bast fiber yield in the 4th year of cultivation. Only a few data (*n* = 3) are reported for spontaneous nettle with a lower yield than cultivated clones [4,11]. The majority of available data (*n* = 15) [1,2,4,11,21,140,151,152,153,154,155,156,157,158] refers to the 2nd year of cultivation and to “Clone 13” (*n* = 5) [2,21,140,154,156]. Due to the wide range of climatic zones and agronomic managements, nettle stem yield, if 1st year is excluded from the analysis, ranges from 1.8 Mg DM ha^−1^ to 15.4 Mg DM ha^−1^.

### 4.3. Fiber Extraction and Processing

Even though the extraction methods used in the past have been very diverse, the general procedure of nettle (bast) fiber extraction can be subdivided into the following five process steps: harvest, retting, decortication, separation, and refining (Figure 8). This section presents an overview of the possible methods in each process step in Figure 8.

Depending on the epoch and if wild nettles or cultivated nettles are harvested, there are two common possibilities: manual or machine harvesting [9,32]. The next process step, retting, is done to separate the fiber bundles located in the sclerenchyma from the lignified core by both bacteria and fungi, which helps to peel off the bark and to extract the fiber bundles more easily. Traditionally, two retting methods were used: dew (field) or water retting [9,32]. Rarer methods are stand retting (during winter), chemical or enzymatic retting [21,33,171]. Quite often, unretted nettle stems (sometimes referred to as “green nettles stems”) are processed: wild nettles are harvested, the bark is either directly peeled off on-site, or the plants are left to dry, and the leaves and stinging needles are peeled off later without performing a specific retting process [29,34,36,172]. Green nettle stems are often used to avoid over-retting. Especially for wild nettles, with varying stem diameters, it is difficult to achieve the same retting grade for all stems. Two ways are generally important for the decortication process (removing the bast/bark from the inner wooden core): manual and machine decortication (e.g., crushing, breaking unit, hammer mill). For separating the single fibers from the fiber bundles, different methods have been developed in the last centuries: cooking solely in water, chemical, and enzymatic treatments, or mechanical separation methods like scutching, comb shaking, coarse separation, or step cleaning (Figure 8). The last process step of refining refers to the fibers’ final use, e.g., if they are used as relatively coarse fiber bundles (e.g., in needle felts) or as fine fiber bundles or single fibers for spinning (e.g., with a spinning wheel or in cotton spinning machinery). Methods range from cooking, beating, or hackling to carding. Theoretically, the methods shown in Figure 8 can be combined in different settings with each other. For example, spinning into fine yarns requires complete retting or enzymatic treatment and decortication, multiple cleaning steps, and fiber opening via separators and carding machines. For fiber use in technical applications, on the other hand, field retting, decortication with hammer mills, and fiber opening may be sufficient under certain circumstances.

While the optimal planting density for fiber nettle was found to range from 2 to 3 plants m^−2^, future research should focus on finding the optimal cover crop mixtures to be undersown in nettle plantations. Considering the height and diameter of nettle stems, which are similar to those of hemp, the main harvesting option is disordered harvesting (cutting, swatting, and baling). To date, there are no reported experiences on the development for fiber nettle of innovative harvesting systems, such as those tested for fiber or multipurpose hemp [173]. There is thus a demand for testing innovative harvesting techniques.

## 5. Nettle as a Multipurpose Crop

This latter section covers the various uses of nettle, as highlighted in Figure 1, detailing the wide range of end products. The recent European research projects allow for its reviviscence while emphasizing its increasing role in polymer composites and phytomanagement issues.

### 5.1. Potential Industrial Uses of Nettle

Nettle has a wide range of applications, in addition to using for fiber production, and the stinging nettle has been the subject of renewed interest linked to the promotion of phytotherapies and biosourced materials [5,9,18,99,174,175,176,177]. All parts of stinging nettle could have the potential to be valued due to their unique properties. Table 3 summarizes the principal product applications of nettles. An innovative application is the production of carbon nanosheets, which are thick graphite sheets (less than 1 nm) that can be used in polymeric nanocomposites as biosensors or catalyst supports, for example [178,179,180]. Carbon nanosheets have been successfully synthesized from stems of stinging nettle. Producing carbon nanosheets from natural materials could improve the purity of materials by removing some contaminants [181].

### 5.2. Fiber-Based Applications

Thanks to their interesting properties plant fibers have an increasing role as reinforcement in polymer composites [207,209]. The fibers of interest from nettles are the bast fibers. Bast fiber is one type of natural plant fiber coming from the sclerenchyma and associated with the phloem, as detailed in Section 3.2. This type of fiber is of interest because they represent a renewable source that can be used for composite materials for the textile, construction, or automobile industries [210]. Their natural abundance, biodegradability, and strength properties made them attractive as reinforcing agents in polymer composite industries [211].

Stinging nettle fiber has been shown to be up to twice as stiff and strong as hemp and superior to flax (Table 4), with an average stiffness and tensile strength properties ranging from 65 GPa [39,74] to 87 GPa [182] and from 740 MPa to 1594 MPa, respectively. The elongation is similar to that of hemp [2,207]. Nettle fibers could be used as a substitute for glass fibers, such as in the manufacturing of composites for the automotive industry or as a replacement for asbestos fibers, where nettle fibers could be superior to flax fibers. Several studies have already highlighted the potential use of nettle fibers as reinforcement material for composites [184,189,212].

The novel finding here is that the tensile properties of nettle fibers equal and even sometimes surpass those of the best industrial flax fibers and can compete with glass fibers.

### 5.3. Recent Application Developments for Nettle Fibers

The current interest in stinging nettle comes from the need to develop environmentally sustainable crops. Another reason is the negative impact of cotton cultivation, which requires herbicides, pesticides, defoliants, and large amounts of water [218]. More environmentally friendly fiber crops have thus been sought in Europe, and the European community stimulates research for alternative natural fibers [18]. Since the end of the 1990s, there has been a renewed interest in stinging nettle, suggested by the emergence of several projects in Germany, but also more recently in Austria, Finland, France, Italy, Lithuania, and Luxembourg (Appendix A). These projects have mainly focused on novel methods for extracting and producing fibers from nettle and its cultivation (e.g., planting density, nutrient requirements, harvesting methods). The current ERA-NET SusCrop NETFIB project (Valorization of nettle fiber grown on marginal lands in an agro-forestry system, “netfib.eu accessed on 1 October 2020”) aims to develop the production of nettle in an innovative agro-forestry system on marginalized land, to feed a sector in plain expansion, alleviating the pressure on agricultural land.

### 5.4. Nettle Use in Phytomanagement Strategies

A potentially valuable synergy for using nettles for fibers is that nettles can be grown as part of a land management strategy for marginal or brownfield land. Phytomanagement is an approach using plants to stabilize or export soil contaminants while limiting the dispersal and risks of soil contaminants [219]. Several processes can be used in phytomanagement, such as phytoextraction, phytostabilization, phytovolatilization, or rhizofiltration [220]. In many pot-based lab-scale experiments, leaf concentrations of many TE in nettle grown on contaminated soils exceed the toxic levels reported by Kabata-Pendias [116] and in Appendix A. This is the case, for instance, for chromium (Cr) [221,222], selenium (Se) [223], fluorine (F) [224], arsenic (As) [225], zinc [221], lead (Pb), nickel (Ni), and cadmium (Cd) [226]. This ability to accumulate TE has been suggested as an effective adsorbent for removing, e.g., Cu^2+^ from aqueous solutions after incomplete incineration. Thus, it would be a useful biosorbent potential for removing TE (e.g., cadmium) from wastewater [227,228]. This capacity remains to be demonstrated for in situ cases. Indeed, in leaves of nettles collected in situ at contaminated sites, and levels of Cd, Cu, Hg, Ni, Pb, Zn [11,229,230,231,232,233], and Mn [233,234] were well below the toxic levels reported by Kabata-Pendias [116]. However, TE levels in nettle roots are usually much higher than those measured in leaves (Appendix A). Obviously, due to the potential metal concentrations, nettles grown on contaminated lands are not suitable for medical purposes or consumption [235].

Including nettles in phytomanagement strategies also brings substantial biodiversity benefits. Recent approaches to optimizing phytomanagement systems point out the importance of assemblages between crops and the spontaneous species [219] to favor the ecological rehabilitation of these marginal lands [236]. *Urtica dioica* is indeed frequently observed spontaneously growing under Salicaceous species [45], including plantations set up in phytomanagement context [43]. The taxonomic and functional diversity of insects related to nettle were comparable to those in the natural environment, highlighting that nettle also acts as a reservoir of insects in the case of a contaminated site.

As part of a phytomanagement project linked with the production of nettle fibers [237], the entomofauna associated with a nettle-poplar agro-system located at a Hg-contaminated landfill was characterized, and the exposition of insects to Hg was determined. When considering insect life traits, the nettle-related insects were primarily exposed to Hg through the food web with significant biomagnification, particularly at the level of secondary predators. Indeed, within the nettle-related food web, the total Hg concentrations increased as follows: nettles < herbivores < predator specialists < predator generalists, with a gap between predator specialists (including Coccinellidae) and predator generalists, suggesting a likely entry of Hg from external biovectors at this level [238]. Unlike insects related to nettle, the fungal biomass of nettle does not appear to be affected by contamination, but the community structure may be [118]. However, in contaminated environments, the addition of nettle residues (aerial parts) leads to a marked increase in the microbial biomass, C, P, N, and ergosterol content, and, therefore, in the fungal biomass of the soil [118].

Including nettle in phytomanagement practices innovatively offers potential sustainability gains, including (i) recovery of natural fiber, (ii) self-sustaining/maintaining ground cover, (iii) its high biodiversity value as a source for varied food webs, (iv) its deterrence value to visitor damage to phytomanagement systems (keeps people to paths), and (v) its tolerance of a wide range of soil conditions, such as TE contaminated soils.

## 6. Conclusions

Nettles are one of the oldest sources of plant fibers used by humankind but largely faded out of commercial use in the 20th century, except as a wartime material. However, nettles have many properties leading to a resurgence of interest in nettle fiber use in the 21st century. Nettles are relatively easy to cultivate, particularly in temperate climates, as long as there is reasonable rainfall. They are a cosmopolitan species naturally occurring in many regions around the world and so support a very wide range of ecological niches, particularly for invertebrates. They are also relatively resilient to pests and diseases. They are perennials, fundamentally reducing the need for sowing and avoiding tillage. Moreover, they can be grown on marginal land, including areas contaminated by organic or inorganic pollutants, and may potentially play a role in the reduction of contaminant risks to human, water, and ecological receptors. The use of nettles can be synergistic with forestry, for example, the production of poplars and nettles. This is also very interesting for remediation as poplars are widely used in phytoremediation systems. Additionally, nettles are edible and contain a wide range of ingredients and nutraceutical compounds. The functionality of nettle cultivation, therefore, includes combinations of the following:Recovery of natural fibers for use in composite materials and technical textilesRecovery of ingredients and nutraceuticalCo-cropping with treesRisk management for sites affected by land contaminationRecovery of use for marginal landSoil improvement and soil carbon sequestrationProviding a wide range of ecological niches for many native species (so supporting wider ecosystem service delivery).

The exploitation of these functions is in its infancy and depends critically on a sound understanding of the fundamental biology of nettles, their structures, and their cultivation. This review collates in-depth the current state of knowledge for nettles and their cultivation as a baseline resource for the further development of the remarkable opportunities that nettles offer us as a resource and for sustainable land management and improvement. Much of this information in this paper is rather old and has hitherto not been collated in a way that allows it to be used by the wider (online) research community. Moving forward, there are several areas where knowledge and techniques for nettle cultivation and use could be further improved to facilitate the multiple opportunities this crop provides. These include:A greater range of nettle fiber compositional information and functionality testing for modern applications (for example, in reinforcement for composite materials)A greater understanding of the variability of these fibers and how this variability impacts their useA greater understanding of climate effects on nettle growth and nettle fiber propertiesA deeper understanding of nutraceutical and ingredient products available from nettles and the potential to deliver these products in parallel with natural fibers from the same harvested biomassGreater functional understanding of the ecological consequences of nettle production and useA greater effort in piloting and demonstrating nettle production and use, and in particular on marginal areas, including those affected by land contaminationA more robust basis for understanding the economic and wider sustainability consequences of nettle production and use and the potential contribution this might have for addressing the current two leading challenges to humankind: climate change and chemical contamination.

## Figures and Tables

**Figure 1 materials-15-04288-f001:**
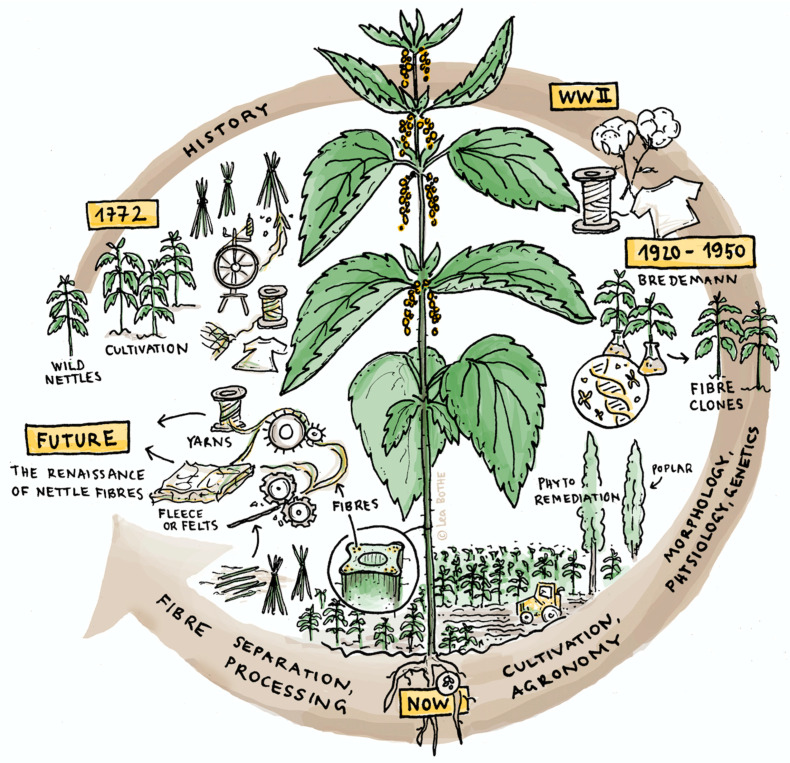
Graphical representation of historical advances in the development of nettle.

**Figure 2 materials-15-04288-f002:**
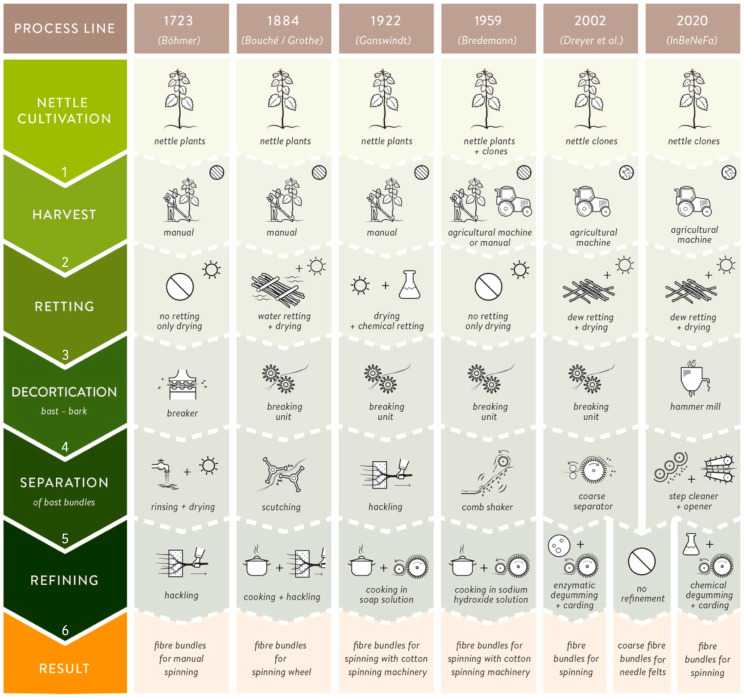
Longitudinal (
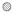
) and disordered (
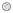
) process lines for nettle fiber extraction and separation from 1723 until present.

**Figure 3 materials-15-04288-f003:**
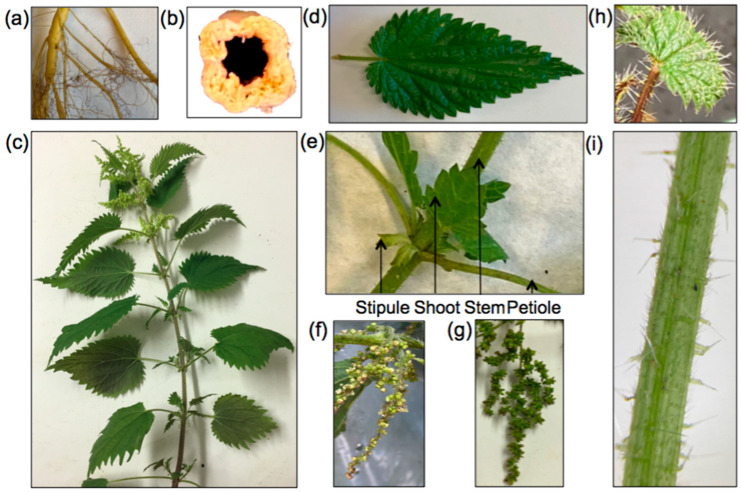
The morphology of the stinging nettle with focuses on (**a**) roots, (**b**) stem section, (**c**) shoots, (**d**) leaf, (**e**) node, (**f**) female and (**g**) male flowers, (**h**) leaf covered with stinging hairs, and (**i**) stinging hairs on the stem.

**Figure 4 materials-15-04288-f004:**
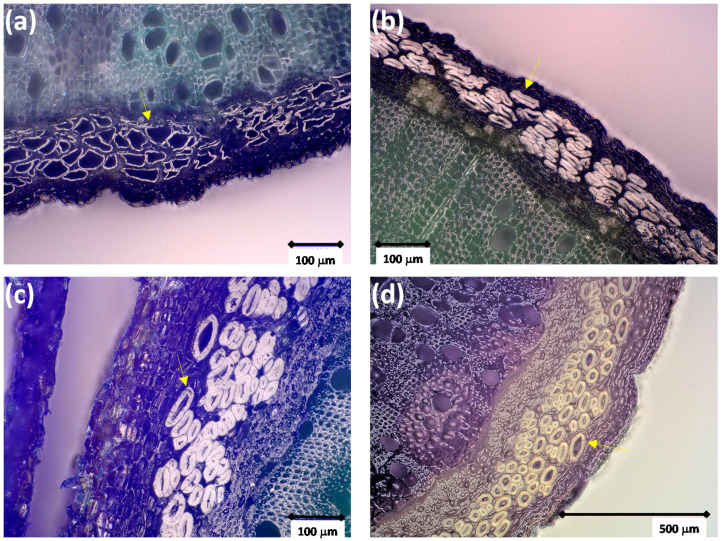
Transverse cross-sections of wild nettle stems harvested in Saint-Symphorien (France, Bourgogne-Franche-Comté region (lat. 47°5′5.98″ N. 5°19′44.0322″ E) in 2019 at different times between May (**a**), June (**b**), July (**c**), and August (**d**). Arrows point out some examples of primary bast fibers (**a**–**c**): Placet et al., unpublished data; (**d**), [11]).

**Figure 5 materials-15-04288-f005:**
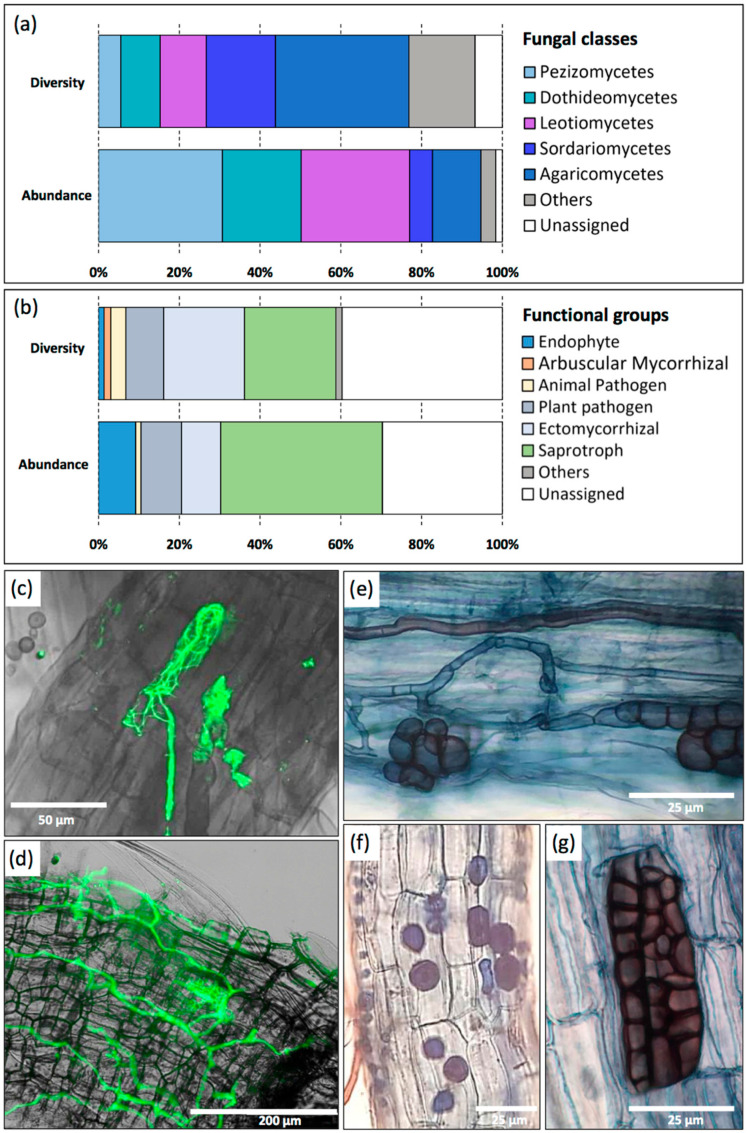
Relative diversity and abundance (%) of the most represented fungal classes (**a**) and functional groups of fungi (**b**) associated with the roots of nettle from the Fresnes-sur-Escaut site. The classes with a relative abundance <5% and the less abundant and diverse functional groups have been gathered in the group “others” (Adapted from [123]). Endophytic fungal structures observed by fluorescence (**c**,**d**) and photonic (**e**–**g**) microscopy in preparations of nettle roots labelled with WGA-AF488 or stained with trypan blue, respectively. (**c**) fungal hyphae colonizing a cortical cell; (**d**) network of extracellular fungal hyphae; (**e**) hyphae forming brain-like microsclerotia; (**f**) intracellular fungal spores with various morphologies; (**g**) melanized fully packed microsclerotia (Yung et al., unpublished data).

**Figure 6 materials-15-04288-f006:**
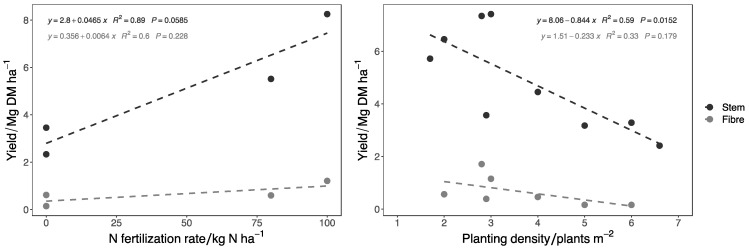
Nettle stem and bast fiber yield (Mg DM ha^−1^) as affected by N fertilization and planting density in published peer-reviewed studies listed in Table 2. Dotted line represents linear regression for the data. Data of spontaneous nettle were excluded from the analyses, as well as mean values representing only 1 study in planting densities higher than 7 plants m^−2^ [150].

**Figure 7 materials-15-04288-f007:**
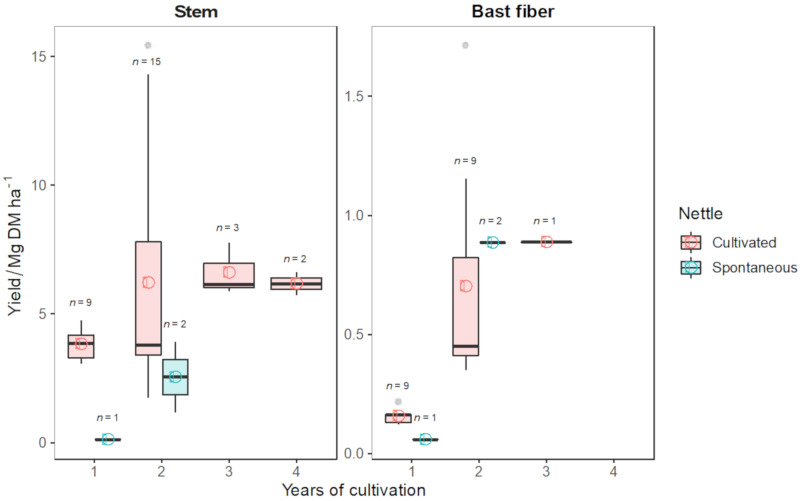
Nettle stem and bast fiber yield (Mg DM ha^−1^) as affected by years of cultivation in spontaneous and cultivated nettle. Points within boxplots are mean values, while the thick line in the boxplot represents the median value. Data have been obtained for studies collected in the peer-reviewed literature (Table 2) [1,2,4,11,21,140,151,152,153,154,155,156,157,158].

**Figure 8 materials-15-04288-f008:**
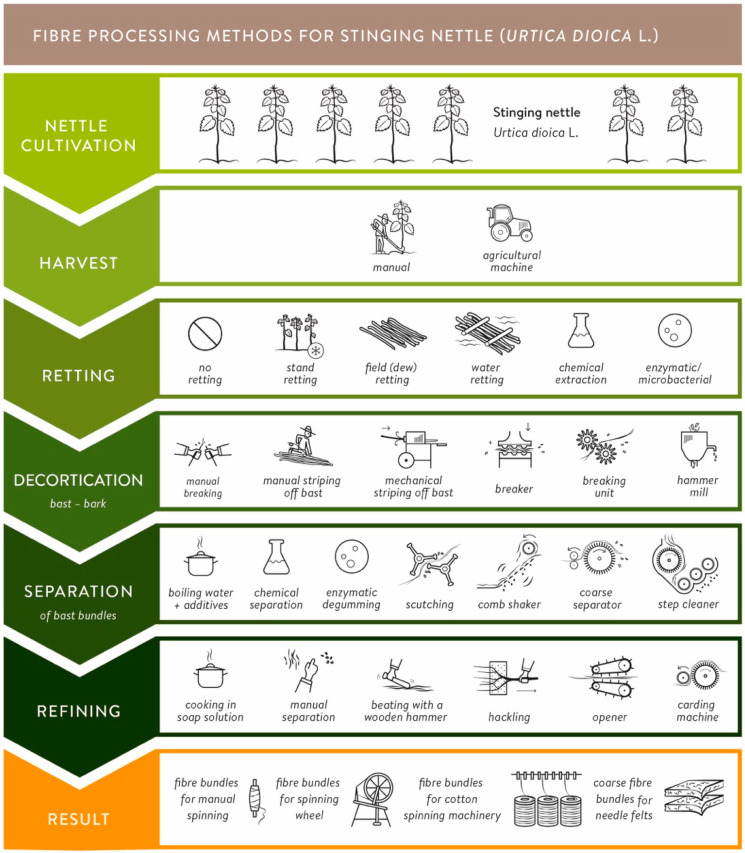
Bast fiber extraction and separation methods for nettle plants (compilation of techniques that graphically illustrate the methods described in the text).

**Table 1 materials-15-04288-t001:** Estimation of *Urtica dioica* genome sizes reported in the literature. The C value expressed in picogram is the mass of the haploid content of a cell. For diploid organisms, each chromosome is present in two copies. To eliminate the redundancy, the final mass is halved or announced as 2C [88]. Genome size (bp) = (0.978 × 10^9^) × amount of DNA (pg) [87]; or 1 pg = 978 Mb [89].

Origin	Genome Size/pg	Haploid Genome Size/Mb	References
Germany	2C = 2.34	572	[90]
Canada	2C = 1.17	572	[91]
Canada	1.20 < 2C < 1.30	611	[92]
UK	1C = 1.6	1564	[93]
Bosnia-Herzegovina	2C = 2.16	528	[94]
Greece to arctic Norway	2C = 1.33 (diploid)	651	[45]
	2C = 2.46 (tetraploid)	602	
Europe and West Asia	2.08 < 2C < 2.20	523	[79]

**Table 2 materials-15-04288-t002:** Summary of data for the studies collected in the literature relative to nettle stem and bast fiber yield (Mg DM ha^−1^) as affected by years of cultivation and agronomic practices (planting density and N fertilization rate).

Reference	Clone	Years of Cultivation	Country	Stem Yield/Mg DM ha^−1^	Bast Fiber Yield/Mg ha^−1^	Planting Density/Plant m^−2^	N Fertilization Rate/kg N ha^−1^
[151]	/	2	Germany	5.85	0.71	4.0	100
[21]	Clone 13	2	Germany	8.20	0.45	2.0	80 *
[152]	/	2	Germany	4.09	0.35	2.9	0
[153]	/	2	Germany	3.80	0.45	/	/
[154]	Clone 13	2	Germany	3.40	/	/	/
[155]	/	2	Germany	7.42	1.16	3.0	80 **
[156]	Clone 13	2	Germany	3.05	0.43	2.9	80 **
[1]	Clone 1-5-7-8-9	2	Austria	3.40	0.36	2.0	80 **
[1]	Clone 1-5-7-8-9	3	Austria	7.78	0.89	2.0	80 **
[140]	Clone 13	2	Germany	3.40	/	2.8	/
[2]	Clone 13	2	Italy	15.42	1.71	2.8	100
[4]	/	2	Lithuania	14.30	/	2.8	32
[4]	/	2	Lithuania	13.61	/	1.7	32
[4]	/	3	Lithuania	6.15	/	2.8	32
[4]	/	3	Lithuania	5.88	/	1.7	32
[4]	/	4	Lithuania	6.63	/	2.8	32
[4]	/	4	Lithuania	5.72	/	1.7	32
[157]	/	1	Greece	3.06	0.22	4.0	0
[157]	/	1	Greece	3.18	0.16	5.0	0
[157]	/	1	Greece	3.29	0.16	6.0	0
[157]	/	1	Greece	3.69	0.16	7.0	0
[157]	/	1	Greece	4.30	0.17	8.0	0
[157]	/	1	Greece	4.76	0.16	9.0	0
[157]	/	1	Greece	3.86	0.12	10.0	0
[157]	/	1	Greece	4.14	0.13	11.0	0
[157]	/	1	Greece	4.18	0.13	12.0	0
[158]	/	2	Croatia	1.75	/	6.6	0
[158]	/	2	Croatia	2.00	/	6.6	50
[158]	/	2	Croatia	3.49	/	6.6	100
[4]	Spontaneous nettle	2	Lithuania	3.92	/	2.8	32
[11]	Spontaneous nettle	1	France	0.12	0.06	/	0
[11]	Spontaneous nettle	2	France	1.17	0.885	/	0

* 20 Mg ha^−1^ of stable manure at the beginning. ** undersown with clovers (*Trifolium incarnatum* or *Trifolium repens*).

**Table 3 materials-15-04288-t003:** Potential end products made from *Urtica dioica* L.

Sector	Use	Part of the Plant	References
Textile/fiber	Clothes, antibacterial finishing of textiles, biobased composites, Carbon nanosheets	Leaves, stem, roots	[147,182,183,184,185,186,187,188,189]
Medicine	Anemia, eczema, antioxidant, analgesic, diabetes, cancer, resistance to bacterial infections	Leaves, stem, roots	[98,190,191,192,193,194,195]
Cosmetics	Soap, shampoo	Leaves, roots	[196,197,198]
Food	Soup, tea, salad, food dye, food additive	Leaves, stem, roots, seeds	[50,199,200,201,202]
Forage crop	Dietary supplements for animals	Leaves, stem, roots	[203,204,205]
Crop farming	Biostimulant, green manure, nettle slurry, pest control, plant-based fertilizer	Leaves, stem, roots	[206,207,208]

**Table 4 materials-15-04288-t004:** Tensile properties of nettle (*Urtica dioica* L.) fibers compared to other European lignocellulosic fibers and usual synthetic fibers.

Type of Fibers		References	Elastic Modulus/GPa	Stress at Failure/MPa	Strain at Failure/%
Nettle (*Urtica dioica* L.)	Range		36–87	711–2196	2.11–2.80
	Mean values and standard deviation of datasets reported in literature	[182]	87 ± 28	1594 ± 640	2.11 ± 0.91
		[213]	79 ± 29	2196 ± 801	2.80 ± 0.90
		[11]	36 ± 19	812 ± 451	2.14 ± 0.81
			53 ± 24	711 ± 427	1.37 ± 0.53
			54 ± 17	1314 ± 552	2.62 ± 1.16
Flax (*Linum Usitatissimum* L.)	Range	[214]	37–75	595–1510	1.60–3.60
	Example of mean values and standard deviation for a dataset	[215,216]	54 ± 15	1339 ± 486	3.27 ± 0.84
Hemp (*Cannabis Sativa*)	Range	[214]	14–44	285–889	0.80–3.30
	Example of mean values and standard deviation for a dataset	[217]	25 ± 11	636 ± 253	2.10 ± 0.70
Glass	Range	[7]	70–85	2000–3700	2.50–5.30
Carbon	Range	[7]	150–500	1300–6300	0.30–2.20

## Data Availability

Not applicable.

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
