# Peer review of "Nettle, a Long-Known Fiber Plant with New Perspectives"

_materials, 2022, doi:10.3390/ma15124288_

Round 1
Reviewer 1 Report
In the manuscript, materials-1731520 authors address the emerging opportunity for nettle fiber production. The manuscript is well-written. Minor corrections are needed before publication.
Hazards associated with nettle cultivation must also be discussed in a separate SubSection of Section 4. Nettle cultivation, harvest, and fiber processing. Exposure to nettle pollen could cause allergy - Tiotiu et al. (2016). Urtica dioica pollen allergy: Clinical, biological, and allergomics analysis. Annals of Allergy, Asthma & Immunology, 117(5), 527-534. Cultivation on marginal land could be associated with health risks due to the accumulation of potentially toxic elements - Bărboiu et al. “Potential health risk assessment associated with heavy metal accumulation in native Urtica dioica.” Romanian Reports in Physics 72 (2020): 711.
Paragraph 306-321 could be moved to this new subsection.
Minor comments.
L85-L88 – Please reformulate the paragraph. in the present form, a reader could understand that Figure 1 refers to the ”various possible value chains and uses of nettle.”
SubSection 3.4. Nettle phytochemistry – please include a paragraph about soluble silicon/silica - Sadowska & Åšwiderski (2020). Sources, Bioavailability, and Safety of Silicon Derived from Foods and Other Sources Added for Nutritional Purposes in Food Supplements and Functional Foods. Applied Sciences, 10(18), 6255.
Table 3, last row - please change from Bioenergy to Crop farming. Biostimulant/nettle slurry/nettle macerated extract and green manure are used not only for bioenergy crops but also in various farming practices. Also, please note that reference 189 is related to the effects against plant-parasitic nematodes. Therefore, please add – pest control to the column Use.
Reviewer 2 Report
Review report
Overall, the topic is interesting and covers a wide range of information regarding Urtica dioica L. Nettle. The plagiarism report is also below 19% However, there are some major weaknesses that must have to be addressed. The literature cited is elaborative of research till 2020 as noted in bibliography, Tables, Figures, and text. The authors have claimed at many places in text about current status of corresponding literature regarding Urtica dioica L. . Therefore, it must have to cover atleast 15% of total references from 2021 and 2022.
Another major weak aspect is that the authors do not conclude different sections with novel findings, which is one of the main requirements for any review to be published in 2022. Furthermore, there are many places where the literature references are not cited.
The abstract includes long sentences that lose the interest of the reader. Rewrite or modify the abstract with short sentences.
Rewrite the conclusions.
Following points are also noted for authors to improve,
Introduction
Line 46 to 72 is also linked to Figure 1. It is suggested to cite the Figure in it.
Line 54: natural or biobased fibers [2] Forecasting studies
Punctuation missing
Lines 59-64
Literature References missing
Lines 64-68
Literature References missing
Lines 68-69
Literature References missing
Line 82: that male and female flowers are located on separate plants [9].
The authors explain the neetles in the early introduction followed by first appearance of Urtica dioica L.. in lines 73 to 82. from there on, the authors jumps into review details.
it is suggested to add the reason for selecting the Urtica dioica L. among neetles family after line 82.
Paragraph 83-91
Make this paragraph a bit more rich in terms of real attraction for future researchers.
Line 121-123
Sentence is not confusing. The brackets are missing.
Line 124:: These clones were re-evaluated in 1993 [18].
What had happened from 1942 to 1993 due to which the research on nettle clones was discontinued?
Line 124-128
Literature References missing
Line 130: tested, ” clone B 13” was
wrong punctuations.
Line 131: experimental field trials [8,18] Francken-
missing punctuations.
Line 135-137
Literature References missing
Line 137-138: It can be assumed that the fiber content was de-137 termined by laboratory-based, mechanical decortication.
What is the need of mentioning method of determining fiber content here?
The method of determined fiber content is not being mentioned in early discussion!
either add similar discussion in early text to maintain the uniformity.
Line 152: years and two location trials with selected
Mention the names of two locations here
Line 166: Figure 2 illustrates the development
Figure 2 includes the formation till 2020. This has gone approximately two years old now. There may be some changes in the "refining" and "separation". either modify 2020-present.
or mention clearly that there is no change after 2020 in any of the schematic contents.
Line 182: Figure 2
Mention Line number in Figure as cited in paragraph (218 to 265)
Line 221: Dreyer et al. [?]…
add reference after et al.
Line 224: by Dreyer et al. [?]is …
add reference after et al.
Line 226: work of Dreyer et al.[?], the production…
add reference after et al.
Line 227: implemented semi-industrially [39].
What is meant by semi industrially?
Line 229: in Figure 2, line 6. Cultivated…….
Mention Line number in Figure
Line 235: in Figure 2, lines 5 and 6) coarse….
Mention Line number in Figure
Lines 237-243
Is it your own work or from literature?
Mention the literature reference here
Section 3.1. Biology, ecology and reproduction 253
Divide this section into three subsection of subheading of 3rd level
3.1.1. Biology
3.1.2. Ecology
3.1.3. Reproduction
if possible.
Lines 254-256
Literature References missing
Line 277-278: Figure 3. The morphology of the stinging nettle with focuses on (a) roots, (b) stem section, 277 (c) shoots (d) leaf, (e) node, (f) female and, (g) male flowers, (h) leaf covered with stinging 278 hairs, and (i) stinging hairs (arrows) on the stem.
Figure 3i is not clear
Line 329: This reproductive strategy may explain the…………
Why do the authors are not sure to cite reproductive strategies?
Lines 342-345:
Literature References missing
Lines 377-379: Figure 4 caption
Literature References missing
Lines 525-532: Figure 5 caption
Missing literature citation
Lines 643-646: Figure 5 caption
Missing literature citation
Lines 659-662
Missing literature citation
Lines 673-674
Missing literature citation
Line 715: other valuable purposes [10].
Explain the valuable purposes here.
Lines 716-725
Missing literature citation
Lines 728-730: Figrue 7 caption
Missing literature citation
Lines 733-739
Missing literature citation
Lines 750-753
Missing literature citation
Lines 773-771: Theoretically, the methods shown in Figure 8 can be combined in different settings with each other.
Explain settings a bit more
Section 5.1
What is the novel information here?
conclude this section to some novel findings
Section 5.2
What is the novel information here?
conclude this section to some novel findings
Section 5.3
What is the novel information here?
conclude this section to some novel findings
Section 5.4
What is the novel information here?
conclude this section to some novel findings
Conclusion
Rewrite the conclusion. The conclusion seems an introduction rather than consolidation of results or set of novel findings into a concise paragraph. Use more than one paragraph or bullets to highlight the findings.

Round 2
Reviewer 2 Report
The authors have included all suggested changes and thus recommended for publication.